# Penalty Virtual Element Method for the 3D Incompressible Flow on Polyhedron Mesh

**DOI:** 10.3390/e24081129

**Published:** 2022-08-15

**Authors:** Lulu Li, Haiyan Su, Yinnian He

**Affiliations:** College of Mathematics and System Sciences, Xinjiang University, Urumqi 830046, China

**Keywords:** virtual element method, 3D incompressible flow, penalty method, polyhedron mesh, error analysis

## Abstract

In this paper, a penalty virtual element method (VEM) on polyhedral mesh for solving the 3D incompressible flow is proposed and analyzed. The remarkable feature of VEM is that it does not require an explicit computation of the trial and test space, thereby bypassing the obstacle of standard finite element discretizations on arbitrary mesh. The velocity and pressure are approximated by the practical significative lowest equal-order virtual element space pair $\left(X_{h},Q_{h}\right)$ which does not satisfy the discrete inf-sup condition. Combined with the penalty method, the error estimation is proved rigorously. Numerical results on the 3D polygonal mesh illustrate the theoretical results and effectiveness of the proposed method.

## 1. Introduction

In computational fluid dynamics, in order to simulate fluid flow, the incompressible fluid equation is numerically solved. There are classical finite element methods [1] and finite volume methods [2]. It is well known that the traditional finite element method or finite volume method are based on the known interpolation. They are mostly used in structural meshes or quadrilateral meshes, the requirements of meshes are coordinated in the implementation process, so the smoothness of interpolation functions is difficult to improve, which greatly limits the numerical accuracy. The newly developed virtual element method (VEM) [3,4,5] benefits from not requiring a construction of the basis function explicitly to avoid these difficulties. The virtual element space can be constructed by the degrees of freedom of the interior and boundary of the element, and the function space does not need to be expressed explicitly. Therefore, it can be applied to any polygon (polyhedron) with convex or non-convex vertices, and the change of the number of nodes is not needed to recalculate the basis function. Thus, the mesh selection bears great flexibility in the calculation [6,7,8,9].

Due to its flexibility in mesh processing and avoidance of explicit construction of shape functions, VEMs have been successfully applied to a large number of problems. For example, conforming and non-conforming VEMs are presented for elliptic problems in [10,11]. Furthermore, H(div)-conforming and H(curl)-conforming virtual elements are introduced in [12].

In the application of practical problems, the virtual meta method has also made outstanding contributions [13,14,15]. These were developed in [16,17,18] for conforming virtual elements for elasticity problems. Conforming and non-conforming VEMs for fourth-order problems were presented in [19,20,21,22]. $C^{1}$-continuous VEMs for the Cahn–Hilliard equation were presented in [23]. For the Stokes problem, several VEMs emerged, such as flow VEM formulations [24], scatter-free virtual elements [25] and non-conforming-required VEMs [26]. When constructing the virtual element space, these methods use more complex degrees of freedom to solve the Stokes equation. For example, in [25], in addition to the degrees of freedom we use, the degrees of freedom also need the moment of the normal vector outside the unit on the edge, and our method is relatively simple. By combining with the penalty method, we can solve the problem of velocity pressure compatibility encountered in the solution of Stokes equation. For the discretization of higher order problems, a class of $H^{\alpha }$-conforming (α>0) elements was proposed in [27] to satisfy arbitrary regularity requirements. VEMs for parabolic and hyperbolic problems were also developed. Moreover, parabolic and hyperbolic problems were developed in [28,29], respectively. A plane-wave virtual element method for the Helmholtz problem was proposed in the literature [30]. In [31], Mascotto studied the behavior of the stiffness matrix resulting from the approximation of the two-dimensional Poisson problem via the virtual element method. In addition, the SUPG stabilization of the virtual element formulation for advection–diffusion problems was presented in [32], and the $h_{p}$ virtual element was studied in [33]. In these references, most of the work focused on conforming VEMs with discrete spaces as subspaces of the original space.

In the finite element approximation, a variety of hybrid finite element algorithms which satisfy the velocity pressure inf-sup condition have been deeply studied. This condition makes more ideal finite element space unavailable, such as the lowest equal-order $P_{1}$-$P_{1}$ finite element pairs. In practical scientific calculation, the equal-order velocity–pressure pairs are very practical owing to their simple data structure, small amount of calculation and high accuracy. In addition, they violate the discrete inf-sup condition, that is, the compatibility between velocity and pressure space, leading to pressure oscillation. In order to compensate for the lack of stability of the inf-sup condition, several stabilized finite element methods for incompressible flows are proposed, such as the penalty method [1], regularization method [34], rich multi-scale method [18], local Gauss integral method [2], and so on. Therefore, it is natural to combine the virtual element method with the penalty method to solve the incompressible flow problem.

The outline of this paper is as follows. In the next section, we provide the theoretical results and related continuous forms of the penalty Stokes equations. Section 3 introduces the virtual element space and the virtual element method. Then, the stability and error results of the penalty virtual element method are given. Finally, the numerical results are provided to verify the theoretical analysis and the effectiveness of the proposed method.

## 2. The Stationary Penalty Stokes Equations

Let Ω be a polygonal domain in $R^{3}$. This paper considers the 3D incompressible Stokes equations to illustrate the penalty virtual element method:(1)$$\begin{array}{c} \left\{\begin{array}{cc} -\Delta u+\nabla p=f, & \mathrm{in}\;\;\Omega ,\\ \nabla \times u=0, & \mathrm{in}\;\;\Omega ,\\ u=0, & \mathrm{on}\;\;\partial \Omega , \end{array}\right. \end{array}$$
where u and *p* are the velocity field and the pressure field, respectively. f represents the external force [35].

The penalty method applied to (Equation 1) is to approximate the solution (u,p) by $\left(u_{\epsilon },p_{\epsilon }\right)$ satisfying the following steady-state penalty Stokes Equation [34]
(2)$$\begin{array}{c} \left\{\begin{array}{cc} -\Delta u_{\epsilon }+\nabla p_{\epsilon }=f, & \mathrm{in}\;\;\Omega ,\\ \nabla \times u_{\epsilon }+\epsilon p_{\epsilon }=0, & \mathrm{in}\;\;\Omega ,\\ u_{\epsilon }=0, & \mathrm{on}\;\;\partial \Omega , \end{array}\right. \end{array}$$
where the penalty parameter is 0<ϵ<1.

Consider the Hilbert spaces
$$\begin{array}{c} V:={\left[H_{0}^{1}\left(\Omega \right)\right]}^{3},\;\;Q:=L_{0}^{2}\left(\Omega \right)=\left\{q\in L^{2}\left(\Omega \right):\;\int_{\Omega }qd\Omega =0\right\}, \end{array}$$
with norms ${\left||v|\right|}_{1}:={\left||v|\right|}_{{\left[H_{0}^{1}\left(\Omega \right)\right]}^{3}}$, ${\left||q|\right|}_{Q}:={\left||q|\right|}_{L^{2}\left(\Omega \right)}$.

The Galerkin variational formula of Equation (Equation 1): find (u,p)∈V×Q, namely
(3)$$\begin{array}{c} \left\{\begin{array}{c} a(u,v)-b(v,p)=(f,v),\;\forall v\in V,\\ b(u,q)=0,\;\forall q)\in Q, \end{array}\right. \end{array}$$
where the bilinear forms a(×;×):V×V→R and b(×;×):V×Q→R be defined by:$$a\left(u,v\right):=\int_{\Omega }\nabla u\nabla vd\Omega ,\forall u,v\in V,$$
$$b\left(v,q\right):=\int_{\Omega }divvqd\Omega ,\forall v\in V,q\in Q.$$

It is well known that (see for [2,35]):a(×,×) and b(×,×) are continuous, i.e.,
$$\left|a\left(u,v\right)\right|\leq {\left||u|\right|}_{1}{\left||v|\right|}_{1}\;\;\forall u,v\in V,$$
$$\left|b\left(v,q\right)\right|\leq {\left||v|\right|}_{1}{\left||q|\right|}_{Q}\;\;\forall v\in V,and\;\;\forall q\in Q,$$a(×,×) is coercive, i.e., there exists a positive constant α such that
$$a\left(v,v\right)\geq {\alpha ||v||}_{1}^{2},\forall v\in V;$$Moreover, the bilinear form b(×,×) satisfies the inf-sup condition: where there exists a constant $\beta_{0}>0$ such that
$$\begin{array}{c} \underset{v\in V}{\sup}\frac{b(v,q)}{{\left||v|\right|}_{1}}\geq \beta_{0}{\left||q|\right|}_{0},\;\forall q\in Q. \end{array}$$

The variational formula of Equation (Equation 2): find $(u_{\epsilon },p_{\epsilon })\in V\times Q$ such that
(4)$$\begin{array}{c} \left\{\begin{array}{c} a\left(u_{\epsilon },v\right)-b\left(v,p_{\epsilon }\right)=\left(f,v\right),\;\forall v\in V\;\\ b\left(u_{\epsilon },q\right)+\epsilon \left(p_{\epsilon },q\right)=0,\;\forall q\in Q. \end{array}\right. \end{array}$$
**Theorem** **1.***There exists at least a solution pair $(u_{\epsilon },p_{\epsilon })\in V\times Q$ which satisfies (Equation 4) and*(5)$$\begin{array}{c} ||\nabla u_{\epsilon }{\left|\right|}_{0}\leq {\left||f|\right|}_{-1}. \end{array}$$*Moveover, if $f\in L^{2}\left(\Omega \right)$, and ϵ satisfies $\epsilon C_{0}\leq 1/2$, then the solution $\left(u_{\epsilon },p_{\epsilon }\right)$ of (Equation 4) satisfies the following regularity:*(6)$$\begin{array}{c} \left|\right|u_{\epsilon }{\left|\right|}_{2}+1/2||p_{\epsilon }{\left|\right|}_{0}\leq C_{0}{∥f∥}_{0} \end{array}$$
where $C_{0}$ denotes some generic constant depending on the data (Ω,f), which may stand for different values at its different occurrences.

There are the following error estimates [1,34].

**Lemma** **1.**
*Assume that $f\in L^{2}\left(\Omega \right)$, (u,p) and $\left(u_{\epsilon },p_{\epsilon }\right)$ are the solution of variational formulas (Equation 3) and (Equation 4), respectively. Then*

$$\begin{array}{c} ||\nabla \left(u-u_{\epsilon }\right){\left|\right|}_{0}+||u-u_{\epsilon }{\left|\right|}_{0}+||p-p_{\epsilon }{\left|\right|}_{0}\leq c\epsilon {∥f∥}_{-1}. \end{array}$$



Please refer to [1,34] for details of certification.

## 3. The Penalty Virtual Element Method for Stokes Equations

The model problem (Equation 1) where Ω is (for simplicity) a convex polyhedron. We suppose that we are given a decomposition $T_{h}$ of Ω in rather general polyhedra *K*. We can assume that $T_{h}$ satisfies the regularity assumption [5]: ∃ a constant C>0 assume that
each polyhedron $K\in T_{h}$ is star-shaped with respect to every point of a ball of radius $\geq Ch_{K}$;for every face f∈∂K we have $h_{f}\geq Ch_{K}$ and *f* is star-shaped with respect to every point of a disk of radius $\geq Ch_{f}$;for every edge e∈∂f, we have $\left|e\right|\geq Ch_{K}$
where $h_{K}$ is the diameter of element *K*, that is, the maximum distance between two nodes of element *K*; *f* is a face of element *K*; $h_{f}$ is the diameter of the face *f*; *e* is the edge length of the element *K*.

For k≥1, the same discussion as in [10] reveals that the dimension of $V_{h}^{K}$ is
$$\begin{array}{c} N_{K}^{V}=\left\{\begin{array}{cc} 2nk+k(k-1), & for\;\;d=2,\\ \frac{3}{2}nk\left(k+1\right)+\frac{1}{2}\left(k-1\right)k\left(k+1\right),\; & for\;\;d=3. \end{array}\right. \end{array}$$

One of the advantages of the virtual element method is that it does not need to display the construction unit basis function, so one of our most important calculation tools is the definition of scale monomials. The form of scale monomials is given below
$$M_{k}:=\{m_{\alpha }=\frac{{\left(x-x_{K}\right)}^{\alpha }}{h_{K}},|\alpha |\leq k\}$$
where x=(x,y,z) is a vector, $\alpha =(\alpha_{1},\alpha_{2},\alpha_{3})$ is a triple index, $h_{K}$ is the element *K* diameter, and $x_{K}=\left(x_{K},y_{K},z_{K}\right)$ is the element *K* centroid.

Different from the construction of the two-dimensional (2D) virtual element space, in the case of 2D, the lower order polynomial of polygon cannot be obtained. Therefore, we calculate on the edge of the element and extend it to the whole space by using the boundary value problem. Therefore, the 2D virtual element space mainly considers the nodes, edges and interior of the element. In the construction of the 3D virtual element space, we should not only consider the nodes and interior of the element, but also the nodes, ridge and interior of the face on each face
$$W_{k}\left(f_{K}\right):=\left\{v\in H^{1}\left(f_{K}\right):v{|}_{\partial f_{K}}\in B_{k}\left(\partial f_{K}\right),\Delta v\in P_{k-2}\left(f_{K}\right)\right\},$$
where $B_{k}\left(\partial f_{K}\right):=\left\{v\in C^{0}\left(\partial f_{K}\right):v{|}_{e}\in P_{k}\left(f_{K}\right),\forall e\subset \partial f_{K}\right\}$. Here $W_{k}\left(f\right)$ is the function space of a face $f_{K}$ defined in an element *K*, $B_{k}\left(\partial f_{K}\right)$ space is defined on the face $f_{K}$, *e* is an edge of the face $f_{K}$, and $\partial f_{K}$ is all the edges of the face $f_{K}$. $P_{k}\left(f_{K}\right)$ is the set of polynomials on $f_{K}$ of degree ≤k.

With the definition of the space of a single face $W_{k}\left(f\right)$, we give the definition of the virtual meta space of all faces
$$W_{k}^{\partial K}\left(K\right):=\left\{v\in C^{0}\left(\partial K\right):v{|}_{f_{K}}\in W_{k}\left(f_{K}\right),\Delta v\in P_{k-2}\left(K\right)\right\},$$
here, ∂K is the set of all faces of element *K*.

The definition of local finite element spaces is given by
$$W_{k}\left(K\right):=\{v\in H^{1}{\left(K\right):v|}_{\partial K}\in V_{k}^{\partial K}\left(K\right),\Delta v{|}_{K}\in P_{k-2}\left(K\right)\}.$$

For each polyhedron *K*, we define a local finite element space $V_{k}\left(K\right)$. Roughly speaking, $V_{k}\left(K\right)$ contains all polynomials of degree *k* (which is essential for convergence) plus other functions whose restriction on a face is still a polynomial of degree *k*.

Note that even here a polynomial of degree *k* satisfies the conditions above, so that $P_{k}\left(K\right)$ is a subspace of $V_{k}\left(K\right)$. We can take the following degrees of freedom in $V_{k}\left(K\right)$:the value of $v_{h}$ at the vertices of *K*;on each edge *e*, the value of $v_{h}$ at the k−1 internal points of the (k+1)-points Gauss–Lobatto quadrature rule on *e*;for each face *f* the moments up to order k−2 of $v_{h}$ in *f*:
$$\int_{f}v_{h}m_{\alpha },\;\;\alpha =1,\ldots ,n_{k-2},$$
where the scaled monomials $m_{\alpha }\in M_{k-2}\left(f\right)$ are defined by
$$m_{\alpha }\left(x\right):={\left(\frac{x-x_{f}}{h_{f}}\right)}^{\alpha };$$the moments up to order k−2 of $v_{h}$ in *K*:
$$\int_{K}v_{h}\mu_{\alpha },\;\;\alpha =1,\ldots ,\nu_{k-2},$$
where the scaled monomials $\mu_{\alpha }\in M_{k-2}\left(K\right)$ are defined in (1.5), and $\nu_{k-2}$ by
$$\nu_{\alpha }\left(x\right):={\left(\frac{x-x_{K}}{h_{K}}\right)}^{\alpha }.$$

For each polyhedron $K\in T_{h}$ and $v_{h}\in W_{h}\left(K\right)$, we can construct $\Pi_{K}^{\nabla }v_{h}:W_{k}\left(K\right)\to P_{k}\left(K\right)$ according to the method in [6], which is defined as follows
$$\begin{array}{c} \left\{\begin{array}{c} {\left(\nabla w_{h},\nabla \left(\Pi_{K}^{\nabla }v_{h}-v_{h}\right)\right)}_{0,K}=0,\;\forall w_{h}\in W_{h},\\ \;P_{0}\left(\Pi_{K}^{\nabla }v_{h}-v_{h}\right)=0, \end{array}\right. \end{array}$$
where, $P_{0}$ is a projection operator of constant functions, that we choose as
(7)$$\begin{array}{c} \begin{array}{c} P_{0}v_{h}:=\frac{1}{N_{K}^{V}}\sum_{i=1}^{N_{K}^{V}}v_{h}\left(V_{i}\right),\;\;for\;k=1, \end{array} \end{array}$$
(8)$$\begin{array}{c} \begin{array}{c} P_{0}v_{h}:=\frac{1}{\left|K\right|}\int_{K}v_{h},\;\;for\;k\geq 2. \end{array} \end{array}$$

**Remark** **1.**
*Note that, integrating by parts, we have again, for every $p_{k}\in P_{k}\left(K\right)$,*

(9)
$$\begin{array}{c} \begin{array}{c} {\left(\nabla p_{k},\nabla v_{h}\right)}_{0,K}=-\int_{K}▵p_{k}v_{h}+\int_{\partial K}\frac{\partial p_{k}}{\partial n}v_{h}. \end{array} \end{array}$$

*Since $▵p_{k}\in P_{k-2}\left(K\right)$, the first term can again (for k>1) be computed using the degrees of freedom (iv). The second term, instead, cannot be computed directly from the degrees of freedom (iii), since on each face f of ∂K, $\frac{\partial p_{k}}{\partial n}$ is in $P_{k-1}\left(f\right)$, but the choice of using $v_{h}{|}_{f}\in W_{k}\left(f\right)$ allows us to compute the moments of order k-1 and k as well. It is known in [5]: When constructing $L^{2}$- projection operator, the shape function on every polygon K is in the space $W_{k}$. For any k≥1 and the degree of freedom of given shape function $w_{h}$, the moments up to order k can be calculated accurately.*


However, there only exists the $L^{2}$-projection on all polyhedrons of degree no more than (k−2) in $W_{k}\left(K\right)$, which is not enough to deal with more complex situations, e.g., the $L^{2}$-error estimates and lower-order terms. To this end, the modified virtual element spaces on every face element *K*.
$${\tilde{W}}_{k}\left(K\right):=\left\{v_{h}\in W_{k}\left(K\right):{\left(q^{*},v_{h}-\Pi_{K}^{\nabla }v_{h}\right)}_{0,K}=0,\forall q^{*}\in M_{k-1}^{*}\cup M_{k}^{*}\right\}.$$

The $L^{2}\left(K\right)$-orthogonal projection $\Pi_{K}^{0}:{\tilde{W}}_{k}\left(K\right)\to P_{0}\left(K\right)$, and for any $w_{h}\in V_{k}\left(K\right)$,
$${\left(w_{h}-\Pi_{K}^{0}w_{h},m_{k}\right)}_{0,K}=0,\forall m_{k}\in M_{k}\left(K\right).$$

For each polyhedron *K*, a local finite element space $V_{k}\left(K\right)$ is defined. Now, the local virtual element spaces for the velocity and pressure are, respectively, introduced by
$$V_{k}\left(K\right):={\left[{\tilde{W}}_{k}\left(K\right)\right]}^{3},\;\;Q_{k}\left(K\right):={\tilde{W}}_{k}\left(K\right),\;\;\forall K\in T_{h}.$$

Then, we can define the global finite element space $V_{h}$ as
$$\begin{array}{cc} V_{h} & :=\{v_{h}\in {\left[H_{0}^{1}\left(\Omega \right)\right]}^{3}:\;v_{{h|}_{K}}\in V_{k}\left(K\right)\;\mathrm{for}\;\mathrm{all}\;K\in T_{h}\},\\ Q_{h} & :=\{q_{h}\in L_{0}^{2}\left(\Omega \right):\;q{|}_{K}\in Q_{k}\;\forall K\in T_{h}\}. \end{array}$$

### 3.1. Constructing the Discrete Matrix

With the definition of $H^{1}$-projection operator, we first define the local discrete version for (Equation 4). The discrete version of $a_{h}^{K}\left(\times ,\times \right)$ is set as
$$a_{h}^{K}\left(u_{h},v_{h}\right)={\left(\nabla \left(\Pi_{K}^{\nabla }u_{h}\right),\nabla \left(\Pi_{K}^{\nabla }v_{h}\right)\right)}_{K}+S_{K}{\left(u_{h},v_{h}\right)}_{K}$$
$$S_{K}{\left(u_{h},v_{h}\right)}_{K}=h_{K}{\left(\nabla \left(u_{h}-\Pi_{K}^{\nabla }u_{h}\right),\nabla \left(v_{h}-\Pi_{K}^{\nabla }v_{h}\right)\right)}_{K}$$
where $S_{K}$ is any symmetric, positive definite and computable bilinear form that guarantees [5]:$$c_{*}a_{K}\left(v_{h},v_{h}\right)\leq S_{K}\left(v_{h},v_{h}\right)\leq c^{*}a_{K}\left(v_{h},v_{h}\right),\forall v_{h}\in ker\left(\Pi_{K}^{\nabla }\right).$$

Then, it is easy to prove that this discrete local bilinear form satisfies the following consistency and stability assumptions [36]:k-compatibility: if $v\in V_{k}\left(K\right),s\in P_{k}\left(K\right)$, have
$$a_{h}^{K}\left(v,s\right)=a_{K}\left(v,s\right);$$Stability: there are two positive constants $\alpha_{*}$ and $\alpha^{*}$ dependent on $h_{K}$ and *K*, have
$$\alpha_{*}a_{K}\left(v,v\right)\leq a_{h}^{K}\left(v,v\right)\leq \alpha^{*}a_{K}\left(v,v\right),\;\;\forall v\in V_{k}\left(K\right);$$Computability: we can know the computability of the discrete bilinear form from Remark 1.

Finally, we define the global approximated bilinear form $a_{h}\left(\times ,\times \right)$ by simply summing the local contributions
$$a_{h}\left(u_{h},v_{h}\right):=\sum_{K\in T_{h}}a_{h}^{K}\left(u_{h},v_{h}\right),\;\;\forall u_{h},v_{h}\in V_{h}.$$

We define projection $\Pi_{K}^{\mathrm{div}}\left(\nabla \times v_{h}\right):{\left(\Pi_{K}^{\mathrm{div}}\left(\nabla \times v_{h}\right),m_{\alpha }\right)}_{K}={\left(\nabla \times v_{h},m_{\alpha }\right)}_{K},\;\;\forall m_{\alpha }\in M\left(K\right)$. The bilinear form $b_{h}^{K}\left(p_{h},v_{h}\right)$ and global contributions $b_{h}\left(p_{h},v_{h}\right)$ are given as
$$\begin{array}{cc} b_{h}^{K}\left(p_{h},v_{h}\right) & =\left(\Pi_{K}^{0}p_{h},\Pi_{K}^{\mathrm{div}}\left(\nabla \times v_{h}\right)\right),\;\;\forall p_{h}\in Q_{h},v_{h}\in V_{h},\\ b_{h}\left(p_{h},v_{h}\right) & :=\sum_{K\in T_{h}}b_{h}^{K}\left(p_{h},v_{h}\right). \end{array}$$

The linear form f(v) on the right-hand side of the variational problem is discrete by $f_{h}$ such that
$$\left(f_{h},v_{h}\right)=\sum_{K\in T_{h}}\left(f,\Pi_{K}^{0}v_{h}\right){)}_{0,K}.$$

The discrete problem (Equation 2) which we can solve is written as: find $u_{h}\in V_{h}$ such that
(10)$$\begin{array}{c} \begin{array}{c} a_{h}\left(u_{h},v_{h}\right)-b_{h}\left(p_{h},v_{h}\right)+b_{h}\left(u_{h},q_{h}\right)+d_{h}\left(p_{h},q_{h}\right)=\left(f_{h},v_{h}\right),\;\forall v_{h}\in V_{h},\;q_{h}\in Q_{h}, \end{array} \end{array}$$
where
$$d_{h}\left(p_{h},q_{h}\right)=\sum_{K\in T_{h}}\epsilon \left(\Pi_{K}^{0}p_{h},\Pi_{K}^{0}q_{h}\right)+\sum_{K\in T_{h}}\epsilon \left(\left(I-\Pi_{K}^{0}\right)p_{h},\left(I-\Pi_{K}^{0}\right)q_{h}\right).$$

The first term in $d_{h}\left(p_{h},q_{h}\right)$ is the penalty term, and the second term is the stable term.

### 3.2. Theoretical Analysis

We begin by proving an approximation result for the virtual local space $V_{h}$. First of all, let us recall a classical result by Scott–Dupont [37].

**Lemma** **2.**
*Let $K\in \tau_{h}$, then for all $u\in {\left[H^{s+1}\left(K\right)\right]}^{3}$ with 0≤s≤k, there exists a polynomial function $u_{\pi }\in {\left[P_{k}\left(K\right)\right]}^{3}$, such that*

(11)
$$\begin{array}{c} ||u-u_{\pi }{\left|\right|}_{0,K}+h_{K}|u-u_{\pi }{|}_{1,K}\leq Ch_{K}^{s+1}{\left|u\right|}_{s+1,K}. \end{array}$$



We have the following proposition

**Proposition** **1.**
*Let $u\in V\cap {\left[H^{s+1}\left(K\right)\right]}^{3}$ with 0≤s≤k. Under the mesh assumptions, there exists a polynomial function $u_{I}\in V_{h}$, such that*

(12)
$$\begin{array}{c} ||u-u_{I}{\left|\right|}_{0,K}+h_{K}|u-u_{I}{|}_{1,K}\leq Ch_{K}^{s+1}{\left|u\right|}_{s+1,D(K)}, \end{array}$$

*where C is a constant independent of h, and D(K) denotes the “diamond” of K, i.e., the union of the polygon in $T_{h}$ intersecting K.*


**Lemma** **3.**
*Given the discrete spaces $V_{h}$ and $Q_{h}$ defined in (2.3) and (2.4), there exists a positive β, independent of h, such that: the bilinear form $b_{h}\left(\times ,\times \right)$ satisfies the discrete inf-sup condition, i.e.,*

(13)
$$\begin{array}{c} \underset{v_{h}\in V_{h}v_{h}\neq 0}{\sup}\frac{b(u_{h},q_{h})}{\left|\right|v_{h}{\left|\right|}_{1}}\geq \beta ||q_{h}{\left|\right|}_{Q}\forall q_{h}\in Q_{h}. \end{array}$$



**Proof.** We only sketch the proof, because it essentially follows the guidelines of Theorem 3.1 in [38]. Since the continuous inf-sup condition (2.6) is fulfilled, it is sufficient to construct a linear operator $\pi_{h}:V\to V_{h}$, satisfying
(14)$$\begin{array}{c} \left\{\begin{array}{c} b\left(\pi_{h}v,q_{h}\right)=b\left(v_{h},q_{h}\right),\;\forall v\in V,\;\forall q_{h}\in Q_{h},\\ \left|\right|\pi_{h}v||\leq C_{\pi }{\left||v|\right|}_{1},\;\forall v\in V, \end{array}\right. \end{array}$$
where $C_{\pi }$ is a positive *h*-independent constant. Given v∈V, using arguments borrowed from [38] and considering the VEM interpolant $v_{I}$ presented in Proposition (4.1), we first construct ${\bar{v}}_{h}\in V_{h}$ such that
$$b\left(v-{\bar{v}}_{h},{\bar{q}}_{h}\right)=0,\;\forall q_{h}\in Q_{h},$$
and
$$||v-{\bar{v}}_{h}{\left|\right|}_{1}\leq {C||v||}_{1}.$$□

Thus, we can directly have the following two lemmas with analogous proof as in [37], which are omitted here.

**Lemma** **4.**
*There exists a positive constant C such that, for all $E\in T_{h}$ and all smooth enough functions φ defined on K, it holds:*

$$||\varphi -\Pi_{K}^{\nabla }\varphi ||\leq Ch_{K}^{s-m}{\left|\varphi \right|}_{s,K},\;s,m\in N,\;m\leq s\leq k+1,\;s\geq 1,$$


$$||\varphi -\Pi_{K}^{0}\varphi ||\leq Ch_{K}^{s-m}{\left|\varphi \right|}_{s,K},\;s,m\in N,\;m\leq s\leq k+1,\;\;\;\;\;\;\;\;\;\;\;$$


$$||\varphi -\varphi_{I}{\left|\right|}_{m,K}\leq Ch_{K}^{s-m}{\left|\varphi \right|}_{s,E},\;m,s\in N,\;m\leq s\leq k+1,\;s\geq 2.$$



Define a norm as $\left||\right|\left(u_{h},p_{h}\right){\left||\right|}_{h}^{2}:=\left|\right|u_{h}{\left|\right|}_{1}^{2}+\left|\right|p_{h}{\left|\right|}_{0}^{2}+d_{h}\left(p_{h},p_{h}\right)$. We have the following theorem.

**Theorem** **2.**
*Assume that the mesh assumption is satisfied and $h=\max\{h_{K}\}$, $\forall K\in T_{h}$. Let $\left(u_{\epsilon },p_{\epsilon }\right)$ and $\left(u_{h},p_{h}\right)$ is solution of (Equation 4) and (Equation 10), respectively. Then, it holds*

$$\begin{array}{cc} \left|\right|u_{\epsilon }-u_{h}{\left|\right|}_{1}+\left|\right|p_{\epsilon }-p_{h}{\left|\right|}_{0} & \leq C(||u_{\epsilon }-u_{I}|{|}_{1}+||p_{\epsilon }-p_{I}|{|}_{0}+||\nabla \times u_{\epsilon }-\Pi_{K}^{\mathrm{div}}\nabla \times u_{\epsilon }|{|}_{0}\\ \; & \;\;\;\;+|u_{\epsilon }-\Pi_{K}^{0}u_{\epsilon }{|}_{1,h}+\left|\right|p_{\epsilon }-\Pi_{K}^{\mathrm{div}}p_{\epsilon }{\left|\right|}_{0}+||f-f_{h}{\left|\right|}_{0}). \end{array}$$



**Proof.** Combined with the definition of projection operator $\Pi_{K}^{\nabla }$, $\Pi_{K}^{0}$, $\Pi_{K}^{div}$. When k=1, $\Pi_{K}^{\nabla }=\Pi_{K}^{0}$, $\Pi_{K}^{div}$ is the component of $\Pi_{K}^{\nabla }$ (references [4,5]). Firstly, we simply deal with $u_{\epsilon }-u_{h}$ and $p_{\epsilon }-p_{h}$,
$$u_{\epsilon }-u_{h}=u_{\epsilon }-u_{I}+u_{I}-u_{h}=\eta_{u}+\xi_{u},$$
$$p_{\epsilon }-p_{h}=p_{\epsilon }-p_{I}+p_{I}-p_{h}=\eta_{p}+\xi_{p}.$$Next, using the above notation and k-consistency, we obtain
(15)$$\begin{array}{cc} a_{h}\left(\xi_{u},v_{h}\right) & =\sum_{K\in T_{h}}a_{h}^{K}\left(u_{I}-u_{h},v_{h}\right)\\ \; & =\sum_{K\in T_{h}}\left[a_{h}^{K}\left(u_{I}-\Pi_{K}^{0}u_{\epsilon },v_{h}\right)+a_{h}^{K}\left(\Pi_{K}^{0}u_{\epsilon },v_{h}\right)\right]-a_{h}\left(u_{h},v_{h}\right)\\ \; & =\sum_{K\in T_{h}}\left[a_{h}^{K}\left(u_{I}-\Pi_{K}^{0}u_{\epsilon },v_{h}\right)+a^{K}\left(u_{\epsilon }-\Pi_{K}^{0}u_{\epsilon },v_{h}\right)\right]+a\left(u_{\epsilon },v_{h}\right)-a_{h}\left(u_{h},v_{h}\right)\\ \; & =-\sum_{K\in T_{h}}a_{h}^{K}\left(\eta_{u},v_{h}\right)+\sum_{K\in T_{h}}a_{h}^{K}\left(u_{\epsilon }-\Pi_{K}^{0}u_{\epsilon },v_{h}\right)\\ \; & \;\;\;\;-\sum_{K\in T_{h}}a^{K}\left(u_{\epsilon }-\Pi_{K}^{0}u_{\epsilon },v_{h}\right)+a\left(u_{\epsilon },v_{h}\right)-a_{h}\left(u_{h},v_{h}\right). \end{array}$$According to Lemmas 3 and 4, we have
(16)$$\begin{array}{cc} -b_{h}\left(v_{h},\xi_{p}\right) & =-\sum_{K\in T_{h}}{\left(\Pi_{K}^{\mathrm{div}}\nabla \times v_{h},\Pi_{K}^{0}\left(p_{I}-p_{h}\right)\right)}_{0,K}\\ \; & =-\sum_{K\in T_{h}}{\left(\Pi_{K}^{\mathrm{div}}\nabla \times v_{h},p_{I}-p_{h}\right)}_{0,K}\\ \; & =-\sum_{K\in T_{h}}{\left(\Pi_{K}^{\mathrm{div}}\nabla \times v_{h},\eta_{p}\right)}_{0,K}+\sum_{K\in T_{h}}{\left(\nabla \times v_{h}-\Pi_{K}^{\mathrm{div}}\nabla \times v_{h},p_{\epsilon }\right)}_{0,K}\\ \; & \;\;\;\;-\left(\nabla \times v_{h},p_{\epsilon }\right)+\sum_{K\in T_{h}}{\left(\Pi_{K}^{\mathrm{div}}\nabla \times v_{h},p_{h}\right)}_{0,K}\\ \; & =-\sum_{K\in T_{h}}{\left(\Pi_{K}^{\mathrm{div}}\nabla \times v_{h},\eta_{p}\right)}_{0,K}+\sum_{K\in T_{h}}{\left(\nabla \times v_{h}-\Pi_{K}^{\mathrm{div}}\nabla \times v_{h},p_{\epsilon }-\Pi_{K}^{\mathrm{div}}p_{\epsilon }\right)}_{0,K}\\ \; & \;\;\;\;-\left(\nabla \times v_{h},p_{\epsilon }\right)+\sum_{K\in T_{h}}{\left(\Pi_{K}^{\mathrm{div}}\nabla \times v_{h},\Pi_{K}^{0}p_{h}\right)}_{0,K}\\ \; & =\sum_{K\in T_{h}}{\left(\Pi_{K}^{\mathrm{div}}\nabla \times v_{h},p_{\epsilon }\right)}_{0,K}+\sum_{K\in T_{h}}{\left(\nabla \times v_{h},p_{\epsilon }-\Pi_{K}^{\mathrm{div}}p_{\epsilon }\right)}_{0,K}\\ \; & \;\;\;\;-b\left(v_{h},p_{\epsilon }\right)+b_{h}\left(v_{h},p_{h}\right). \end{array}$$Combined with Equation (Equation 16), we can obtain the estimation formula of $b_{h}\left(\xi_{u},q_{h}\right)$
(17)$$\begin{array}{cc} b_{h}\left(\xi_{u},q_{h}\right) & =-\sum_{K\in T_{h}}{\left(\nabla \times \eta_{u},\Pi_{K}^{\mathrm{div}}q_{h}\right)}_{0,K}+b\left(u_{\epsilon },q_{h}\right)\\ \; & \;\;\;\;-\sum_{K\in T_{h}}{\left(\nabla \times u_{\epsilon }-\Pi_{K}^{\mathrm{div}}\nabla \times u_{\epsilon },q_{h}\right)}_{0,K}-b_{h}\left(u_{h},q_{h}\right). \end{array}$$Similarly, we can do the same with $d_{h}\left(\xi_{p},q_{h}\right)$
(18)$$\begin{array}{c} d_{h}\left(\xi_{p},q_{h}\right)=d_{h}\left(p_{I},q_{h}\right)-d_{h}\left(p_{h},q_{h}\right). \end{array}$$Add Equations (Equation 15)–(Equation 18), combine Lemma 2 and Proposition 1 in [4], we have
$$\begin{array}{cc} \; & a_{h}\left(\xi_{u},v_{h}\right)-b_{h}\left(v_{h},\xi_{p}\right)+b_{h}\left(\xi_{u},q_{h}\right)+d_{h}\left(\xi_{p},q_{h}\right)\\ \; & =-\sum_{K\in T_{h}}a_{h}^{K}\left(\eta_{u},v_{h}\right)+\sum_{K\in T_{h}}a_{h}^{K}\left(u_{\epsilon }-\Pi_{K}^{0}u_{\epsilon },v_{h}\right)-\sum_{K\in T_{h}}a^{K}\left(u_{\epsilon }-\Pi_{K}^{0}u_{\epsilon },v_{h}\right)\\ \; & \;\;\;\;+\sum_{K\in T_{h}}{\left(\Pi_{K}^{\mathrm{div}}\nabla \times v_{h},p_{\epsilon }\right)}_{0,K}+\left(f-f_{h},v_{h}\right)+\sum_{K\in T_{h}}{\left(\nabla \times v_{h},p_{\epsilon }-\Pi_{K}^{\mathrm{div}}p_{\epsilon }\right)}_{0,K}\\ \; & \;\;\;\;-\sum_{K\in T_{h}}{\left(\nabla \times \eta_{u},\Pi_{K}^{\mathrm{div}}q_{h}\right)}_{0,K}-\sum_{K\in T_{h}}{\left(\nabla \times u_{\epsilon }-\Pi_{K}^{\mathrm{div}}\nabla \times u_{\epsilon },q_{h}\right)}_{0,K}+d_{h}\left(p_{I},q_{h}\right)\\ \; & \leq C(||\eta_{u}|{|}_{1}+||\eta_{p}|{|}_{0}+||\nabla \times u_{\epsilon }-\Pi_{K}^{\mathrm{div}}\nabla \times u_{\epsilon }|{|}_{0}+|u_{\epsilon }-\Pi_{K}^{0}u_{\epsilon }{|}_{1,h}\\ \; & \;\;\;\;+\left|\right|p_{\epsilon }-\Pi_{K}^{\mathrm{div}}p_{\epsilon }{\left|\right|}_{0}+||f-f_{h}{\left|\right|}_{0})|||\left(v_{h},q_{h}\right){\left||\right|}_{h}. \end{array}$$Thirdly, apply the properties of the defined projection operator to obtain
(19)$$\begin{array}{cc} \left||\right|\left(\xi_{u},\xi_{p}\right){\left||\right|}_{h}\leq & C(||\eta_{u}|{|}_{1}+||\eta_{p}|{|}_{0}+||\nabla \times u_{\epsilon }-\Pi_{K}^{\mathrm{div}}\nabla \times u_{\epsilon }|{|}_{0}\\ \; & +|u_{\epsilon }-\Pi_{K}^{0}u_{\epsilon }{|}_{1,h}+\left|\right|p_{\epsilon }-\Pi_{K}^{\mathrm{div}}p_{\epsilon }{\left|\right|}_{0}+||f-f_{h}{\left|\right|}_{0}), \end{array}$$
combining (Equation 19) and applications of the triangle inequality, this proof is completed. □

## 4. Numerical Experiments

In this section, we will present the results of three numerical experiments to illustrate some of the characteristics of the methods discussed in the previous sections. In the first experiment, we used arbitrary hexahedral mesh to verify the effectiveness and convergence of the method. In the second numerical experiment, we used mesh generation with suspension points and polyhedral mesh. In the third experiment, we considered square cavity flow without true solution. In addition, the meshes we use are divided in the cube area of Ω=[0,1]×[0,1]×[0,1].

We use projectors $\Pi_{K}^{0}$ and $\Pi_{K}^{\nabla }$ to evaluate the error:$L^{2}$-norm: $u_{L^{2}}e_{r}=\sqrt{\sum_{K\in T_{h}}||u-\Pi_{K}^{0}u_{h}{\left|\right|}_{0,K}^{2}}$;$H^{1}$-norm: $u_{H^{1}}e_{r}=\sqrt{\sum_{K\in T_{h}}{\left|u-\Pi_{K}^{\nabla }u_{h}\right|}_{1,K}^{2}}$.

### 4.1. Smooth Solution

We consider the parabolic Equation (Equation 1), where the load term f and the Dirichlet boundary are chosen according to the exact solution $u\left(x,y,z\right)=\left(u_{1},u_{2},u_{3}\right)$ with $\epsilon =O(h_{K})$
$$\begin{array}{c} \left\{\begin{array}{c} u_{1}\left(x,y,z\right)=\sin\left(\pi y\left(y-1\right)\right)\sin\left(\pi z\left(z-1\right)\right),\;\;\;\;\;\;\;\;\;\;\;\;\;\;\;\;\;\;\;\;\;\;\;\\ u_{2}\left(x,y,z\right)=2\left(1-y^{2}\right)y^{2}\left(1-2z\right)\left(1-z\right)z\sin\left(\pi x\left(x-1\right)\right),\;\;\\ u_{3}\left(x,y,z\right)=-2\left(1-z^{2}\right)z^{2}\left(1-2y\right)\left(1-y\right)y\sin\left(\pi x\left(x-1\right)\right),\\ p\left(x,y,z\right)=3-\left(x^{3}+y^{3}+z^{3}\right).\;\;\;\;\;\;\;\;\;\;\;\;\;\;\;\;\;\;\;\;\;\;\;\;\;\;\;\;\;\;\;\;\;\;\;\;\;\;\; \end{array}\right. \end{array}$$

The corresponding results are shown in Table 1, where we use arbitrary hexahedron mesh and Kershaw mesh in Figure 1. Table 1 shows the $u_{H_{1}err}$ and $u_{L_{2}err}$ and $p_{L_{2}err}$ when $\epsilon =O(h_{K})$. It can be seen from the table that the $u_{H_{1}err}$ and $p_{L_{2}err}$ can reach the order 1, and the $u_{L_{2}err}$ can reach the order 2, which is consistent with our theoretical results.

### 4.2. True Solution

Let us provide a numerical example. The solution domain Ω is set as ${\left[0,1\right]}^{3}$. The exact solution u for the velocity is generated by curlΨ, where
$$\Psi =10x^{2}y^{2}z^{2}{\left(x-1\right)}^{2}{\left(y-1\right)}^{2}{\left(z-1\right)}^{2}\left(\begin{array}{c} e^{x+y+z}\\ 100sin(xyz)\\ -10(x^{2}+y^{2}+z^{2}) \end{array}\right),$$
and the pressure is given by
p=−sin(2πx)sin(2πy)sin(2πz).

Through this numerical experiment, we can see from Table 2 that our method is still valid for mesh with hanging points Figure 2a. In addition, for twist polyhedral mesh Figure 2b, our method can reach the theoretical order.

### 4.3. Driven Cavity Flow

We employ a cubic cavity flow to illustrate the feasibility of the 3D Stokes flows. Consider a cube per unit volume here. In this numerical example, the unit tangential velocity in the *x*-direction is prescribed on the top surface (z=1), and u=0 is prescribed on the remaining bounding surfaces, see Figure 3. The experiments of driven cavity flow are mainly carried out on locally refinement mesh to verify that our method is effective for mesh with hanging points (see Figure 4).

Figure 5 shows the velocity cross-section of the first locally refined mesh when the degrees of freedom are 1881 and 13,073, and y=0.5. Figure 6 shows the velocity cross-section of the second locally refined mesh when y=0.5 with 1333 and 9097 degrees of freedom. On the other hand, Figure 7 and Figure 8, respectively, show the velocity vectors of Stokes flow in the *X*–*Y* plane with z=0.5 in the two grid cubic cavities. The velocity affects the distribution of the intensity of the vorticity, so we look more closely at the vorticity profile for more cross-sections.

It can be seen that when the degrees of freedom are similar, the cavity flow phenomena under different grids are similar. In the same grid, the phenomena of different degrees of freedom are stable, which shows that our method is effective.

## 5. Conclusions

In this work, an interesting combination of VEM and the lowest order element with practical significance is proposed for the three-dimensional incompressible flow on polyhedral meshes. In order to better illustrate the method, the optimal error estimate is strictly given by taking the Stokes equation as an example. Numerical experiments verify the theoretical analysis and the effectiveness of the method for arbitrary polyhedral meshes, meshes with hanging points and distorted polyhedral meshes. In our ongoing work, we will consider the adaptive scheme in the VEM framework.

## Figures and Tables

**Figure 1 entropy-24-01129-f001:**
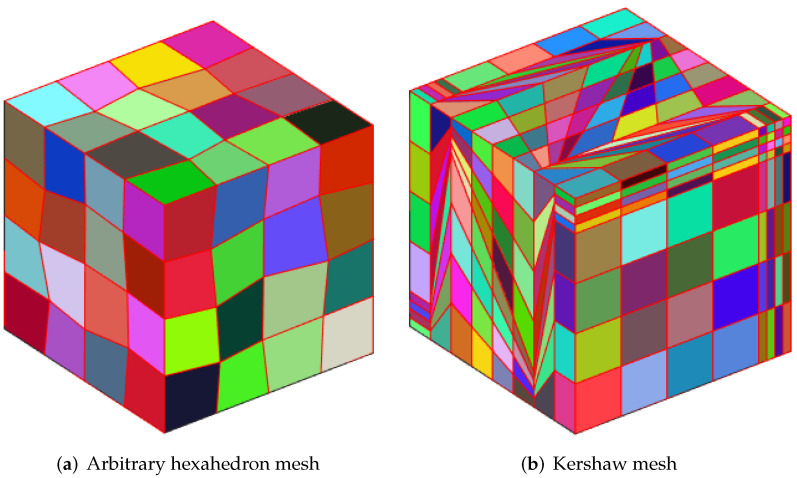
Polyhedral mesh.

**Figure 2 entropy-24-01129-f002:**
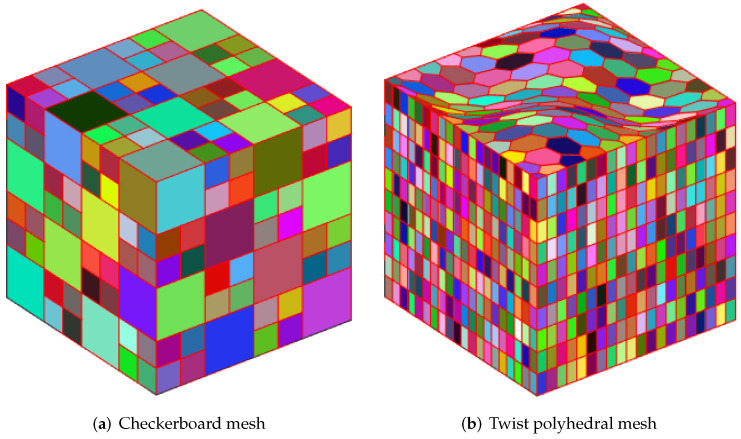
Polyhedral mesh.

**Figure 3 entropy-24-01129-f003:**
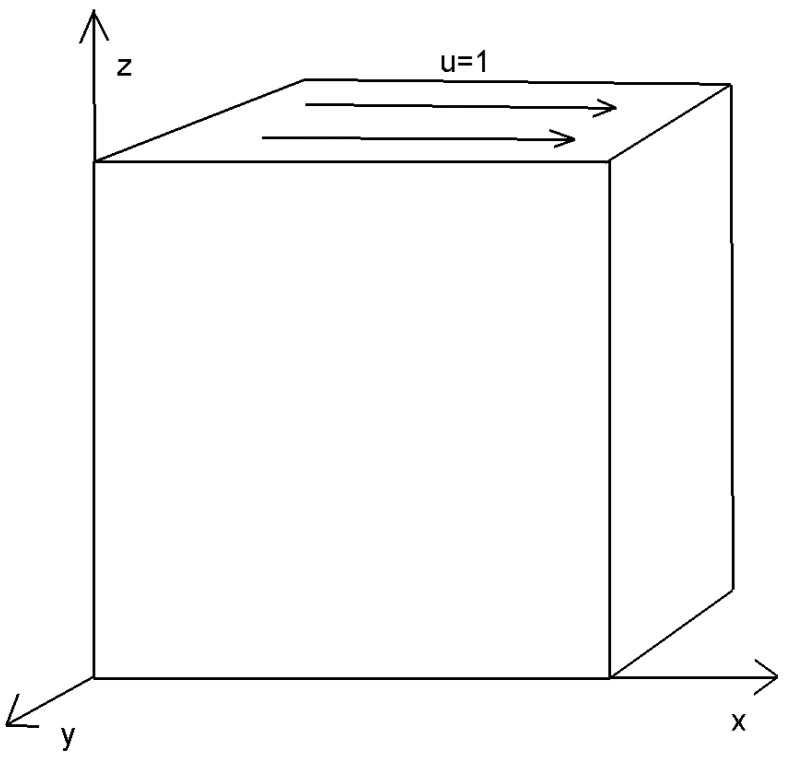
Cubic cavity flow problem with boundary conditions.

**Figure 4 entropy-24-01129-f004:**
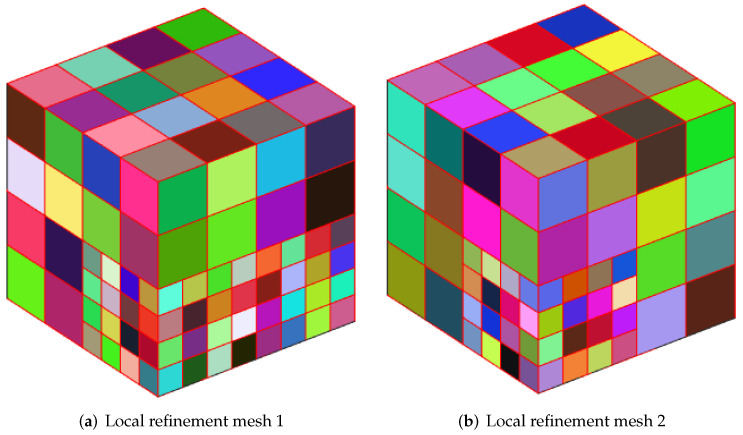
Polyhedral mesh.

**Figure 5 entropy-24-01129-f005:**
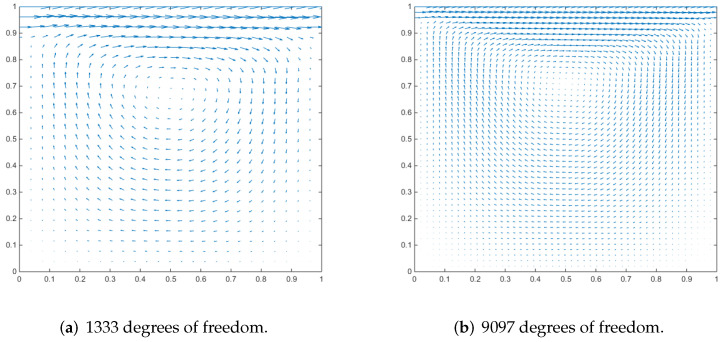
Velocity vectors of x–z plane at y = 0.5 for Stokes flow in a cubic cavity.

**Figure 6 entropy-24-01129-f006:**
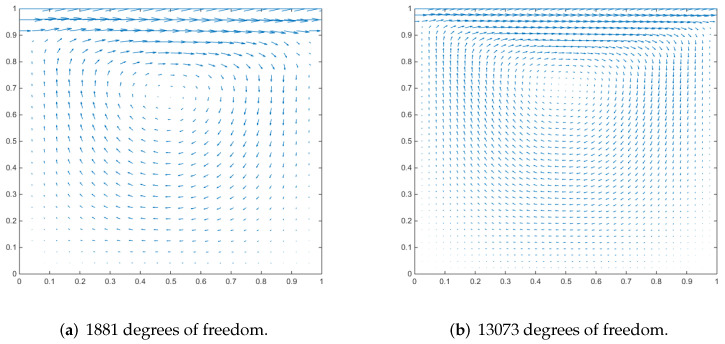
Velocity vectors of x–z plane at y = 0.5 for Stokes flow in a cubic cavity.

**Figure 7 entropy-24-01129-f007:**
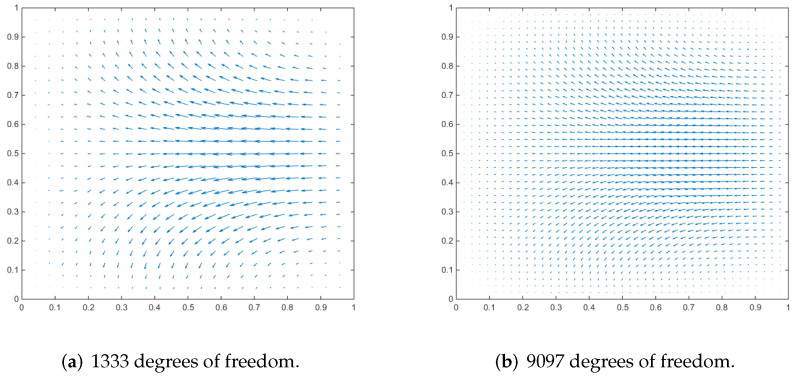
Velocity vectors of x–y plane at z = 0.5 for Stokes flow in a cubic cavity.

**Figure 8 entropy-24-01129-f008:**
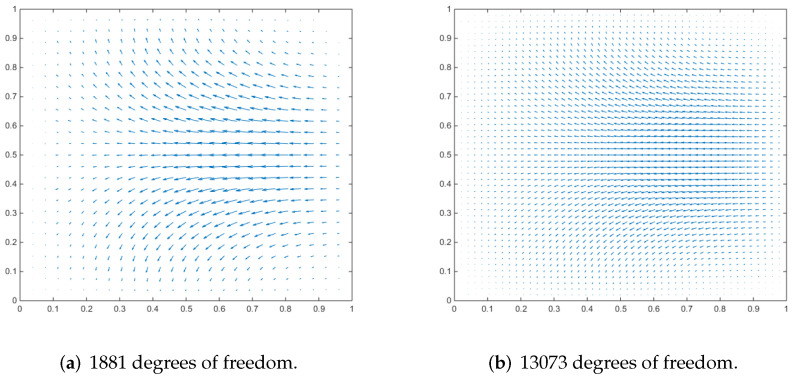
Velocity vectors of x–y plane at z = 0.5 for Stokes flow in a cubic cavity.

**Table 1 entropy-24-01129-t001:** Error results with arbitrary hexahedron mesh and Kershaw mesh.

DOF	$u_{H_{1}}e_{r}$	$u_{H_{1}}e_{r}\mathrm{rate}$	$u_{L_{2}}e_{r}$	$u_{L_{2}}e_{r}\mathrm{rate}$	$p_{L_{2}}e_{r}$	$p_{L_{2}}e_{r}\mathrm{rate}$
729	1.2090 × 10${}^{-1}$	-	8.0410 × 10${}^{-3}$	-	1.1038 × 10${}^{-1}$	-
4913	6.3291 × 10${}^{-2}$	1.0177	2.2149 × 10${}^{-3}$	2.0273	5.6913 × 10${}^{-2}$	1.0471
35,937	3.1722 × 10${}^{-2}$	1.0414	5.7288 × 10${}^{-4}$	2.0387	2.7656 × 10${}^{-2}$	1.0827
729	6.3351 × 10${}^{-1}$	-	2.3690 × 10${}^{-2}$	-	8.2720 × 10${}^{-2}$	-
4913	3.0385 × 10${}^{-1}$	1.1553	6.9287 × 10${}^{-3}$	1.9330	4.8572 × 10${}^{-2}$	0.8371
35,937	1.4716 × 10${}^{-1}$	1.0930	1.8020 × 10${}^{-3}$	2.0304	2.5461 × 10${}^{-2}$	0.9737

**Table 2 entropy-24-01129-t002:** Error results with checkerboard mesh and twist polyhedral mesh.

DOF	$u_{H_{1}}e_{r}$	$u_{H_{1}}e_{r}\mathrm{rate}$	$u_{L_{2}}e_{r}$	$u_{L_{2}}e_{r}\mathrm{rate}$	$p_{L_{2}}e_{r}$	$p_{L_{2}}e_{r}\mathrm{rate}$
625	2.1395× 10${}^{-1}$	-	1.0443× 10${}^{-1}$	-	3.3572× 10${}^{-1}$	-
4417	1.1618× 10${}^{-1}$	0.9368	3.1281× 10${}^{-2}$	1.8494	1.8848× 10${}^{-2}$	0.8857
33,025	6.1371× 10${}^{-2}$	0.9517	3.0917× 10${}^{-3}$	1.9741	8.9848× 10${}^{-3}$	0.9878
3080	6.6406× 10${}^{-2}$	-	2.1750× 10${}^{-2}$	-	1.6897× 10${}^{-2}$	-
20,160	3.3853× 10${}^{-2}$	1.0759	6.0897× 10${}^{-3}$	2.0327	8.6744× 10${}^{-3}$	1.0647
63,240	2.2962× 10${}^{-2}$	1.0187	2.9105× 10${}^{-3}$	1.9373	5.8729× 10${}^{-4}$	1.0235

## Data Availability

Not applicable.

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
