# Peer review of "Penalty Virtual Element Method for the 3D Incompressible Flow on Polyhedron Mesh"

_entropy, 2022, doi:10.3390/e24081129_

Round 1

Reviewer 1 Report

I think the authors should explain in detail what's the difference between their methods and analysis and the other VEMs for the Stokes problem which that just mentioned in the introduction:"For the Stokes problem, several VEMs emerged, such as 31 flow VEM formulations [25], scatter-free virtual elements [26] and non-conforming required 32 VEMs [27,28]." on page 1.

Some sentences are not clear e.g.:

line 78: what does it mean: "The penalty method applied to (1) is to use the solution (u, p) by (uϵ, pϵ) .." use the solution? 

What does it mean here?

line 85: Lemma 1 seems classical.I guess one can refer to a prof instead of giving one. (Or to a similar result)

line 100 (last line) the definition of B_k(\partial f_K) is completely unlear... what is is f_e? what is e? an edge? - called untypically a ridge elsewhere)? What is P(f)? IN nomenclature (cf 57) P is the pressure.. I guess it is something different? The space of constants?

Actually, I have to stop here - it is unreadable - there may be one or two typos on a page but not in a line...

Reviewer 2 Report

This paper addresses a novel framework to solve 3D incompressible flow problems, combining the virtual element method with the penalty method, in order to solve the incompressible flow problems.

The novelty resides in developing a penalty virtual element method which requires considerable development and implementation effort.

The authors explain in a very clear manner the mathematical background. The algorithm is tested on  unsteady Stokes equations model and provides good results. An error–performance test is conducted by varying different model parameters and mesh types. The results are clearly presented.

I strongly recommend the publication only after the following revision recommendations are fulfilled.

Main concerns:

1.Please explain if the trial and test functions coincide or not in your experiments.

2.Please address one of the main problem encountered in the VEM: how did you solve the issue when computing the local stiffness matrix and one of the two entries is a polynomial of degree less than k? Or explain how you avoid this situation.

3.Better emphasize how your technique extracts the right degrees of freedom to produce exact results with right order of magnitude and stability properties.

4. The authors cited an appropriate range of sources. I recommend to mention in Introduction some recent modern alternative methods to solve fluid dynamics problems, i.e. randomized dynamic mode decomposition [1,2], proper orthogonal decomposition [3].

[1] Bistrian D.A., High-Fidelity Digital Twin Data Models by Randomized Dynamic Mode Decomposition and Deep Learning with Applications in Fluid Dynamics. Modelling. 2022; 3(3):314-332. https://doi.org/10.3390/modelling3030020

[2] Bistrian D.A., Navon I.M., Efficiency of randomised dynamic mode decomposition for reduced order modelling. International Journal of Computational Fluid Dynamics. 2018; 32(2-3):88-103.

https://doi.org/10.1080/10618562.2018.1511049

[3] Bistrian D.A., Navon I.M., An improved algorithm for the shallow water equations model reduction: Dynamic Mode Decomposition vs POD. International Journal for Numerical Methods in Fluids. 2015; 78(9):552-580. https://doi.org/10.1002/fld.4029

Minor Typos:

1.Line 16: Explain the acronym VEM at its first use.

2.Line 79: Rephrase : “The penalty method applied to (1) is to use the solution (u, p) by (uϵ, pϵ) satisfies the following steady-state penalty Stokes equations [36].”

- remove the dot from the end of above sentence

-replace the dot from the end of Equation (2) with comma “,”

3.Line 98: Remove “As we all know” at the beginning of the sentence. The readers may be from different domains.

4.Above Line 101: Remove the dot from the end of the sentence “Therefore, we first give the construction of the following space”

5.Line 111: Explain the meaning of notation (iv), (iii) or correct it in the following text:

“the first term can again (for k > 1) be computed using (iv) of the degrees…”; “cannot be computed directly from (iii) of the degrees of freedom…”

6.Line141: Replace “seasons” with “sections”

“…of the methods discussed in the previous seasons.”

7.Line147: Replace “to evaluated” with “to evaluate”

Round 2

Reviewer 2 Report

The authors managed to revise the paper taking into account all the suggestions. I consider that the presentation of the paper is improved and I propose its acceptance in its current form.

Author Response

Thank you very much for your helpful comments and advice concerning our manuscript. Those comments are all valuable and very helpful for revising and improving our manuscript again, as well as the important guiding significance to our researches.